# Stochastic Network Calculus-Aided Delay Analysis of Wireless-Power Line Mixed Networks

**Zheng Li** [1]**, Haiming Hong** [1]**, Lin Pang** [1]**, Muyu Mei** [2,3] **and Qinghai Yang** [2,3,*]

[1] State Grid Laboratory of Power Line Communication Application Technology, Shenzhen Guodian Technology Communication Co., Ltd., Shenzhen 518028, China; lizheng3@sgitg.sgcc.com.cn (Z.L.); honghaimin@sgitg.sgcc.com.cn (H.H.); panglin@sgitg.sgcc.com.cn (L.P.)

[2] State Key Laboratory of ISN, School of Telecommunications Engineering, Xidian University, 2 Taibainan-lu, Xi'an 710071, China; mei_muyu@163.com

[3] Lab of Industrial Internet, Guangzhou Institute of Technology, Xidian University, Guangzhou 510555, China

[*] Correspondence: qhyang@xidian.edu.cn

**Abstract:** In this paper, we investigate the delay performance of a wireless-power line mixed network via a stochastic network calculus (SNC)-based approach. The data transmission in this mixed network is modeled by a two-stage tandem queue, wherein the data is first relayed through a wireless fading channel and then transmitted over a power line communication (PLC) system. The Rayleigh fading captures the wireless fading channel; whereas, the PLC channel gain is characterized by the log-normal distribution. The statistical characteristics of the service processes of both the wireless channel and PLC channel are derived. With any given traffic arrival and the service capability derived, the delay can be easily bounded via SNC.

**Keywords:** stochastic network calculus; power line communications; delay

## 1. Introduction

Due to the rapidly availability and huge geographical area coverage of power lines, power line communication (PLC) has been widely utilized to meet the ever-increasing user demand for speedy access to data [1]. For example, PLC has been applied for smart grid communication from transmission and distribution networks to consumer-end home area networks, for enabling the local area network and for home automation using indoor wiring infrastructure [2].

However, different from conventional wired communication, PLC has its own set of challenges, since the power lines were not initially designed for communication purposes. In particular, the channel fading, noise power and attenuation in PLC randomly fluctuate as the location, time and frequency vary. Thus, it remains a great challenge to integrate PLC into modern communication system and to model PLC channels.

Additive noise in PLC systems is more complex than that in wired communications since it is a mixture of background noise and impulsive noise, which can be described by the Nakagami-m distribution and middleton class A distribution [3]. In addition to these two noise models, Bernoulli–Gaussian distributions are widely utilized to model both types of noises in the literature. Similar to conventional wired and wireless communications, the PLC channel is also affected by multiplicative noise due to the adverse influence of multipath and mismatches in the impedances at the joints. These effects can be well imitated by Rician and Rayleigh fading models [4]. Furthermore, the log-normal distribution has been proven to be an excellent model for PLC communications by the measurement campaign conducted in US urban and sub-urban areas over medium voltage power lines across frequencies of 1.8 to 30 MHz [5].

The PLC channel model has been thoroughly investigated, and the performance of PLC system has been well analyzed over recent decades. The attention and the focus of

researchers has become the wireless-power line mixed communications. It has been proven that network performance can be sufficiently improved when we use a wireless-power line diversity system by employing diverse combinations of the received wireless and PLC signals [6].

The authors of [7] investigated a PLC-assisted wireless relay network wherein power lines initialize and synchronize wireless amplify-and-forward relays and broadcast data between these relays. Similarly, a wireless-PLC relay network was considered in [8] where the wireless channel is occupied by the master node while the PLC channel is occupied by slave nodes.

The standards set by INSTEON Technologies combine wireless and PLC technologies together by utilizing the wireless channel to relay control commands [9]. Furthermore, the physical layer performance can be strengthened with emerging technologies of modern wireless networks, such as beamforming and 5G.

The performance of PLC networks and PLC-wireless networks has been well-investigated. In [10], the bit error rate (BER) performance of OFDM PLC considering the impulsive noise was demonstrated, and the results showed that the OFDM-based PLC systems were subject to much higher error rates in the presence of impulsive noise. Furthermore, the BER performance for a medium-voltage MIMO-OFDM PLC system was studied in [11]. In [12], the trade-off between the algorithm complexity and symbol error rate (SER) was studied in the case of interfacing between wired and wireless communications. Most of the performance analysis works focused on modeling the channel model and noise model for PLC systems and concentrated on BER and SER metrics [13,14].

Though several wireless-PLC mixed networks have been investigated, few works have dealt with the delay analysis problem thus far. To address this problem, we not only need to evaluate the delay for wireless transmission and PLC links but also have to consider that the output of the wireless link is the input of the PLC link.

Recently, stochastic network calculus (SNC) has been proposed to provide a statistical delay envelop for communication systems [15]. SNC in the signal-to-noise-ratio (SNR) domain has been further proposed to capture the service capability of wireless channels, wherein the logarithm operator in the Shannon capacity makes it difficult to express the statistics of the service process in a closed form [16].

In addition, SNC in the SNR domain can also be utilized to analyze the performance for other scenarios, e.g., sensing, controlling and computing, due to the well-known Chernoff's inequality. Last but not least, the convolution operator in the SNC approach makes it possible to deal with the delay analysis problem for tandem queues, which makes it convenient for us to evaluate the performance for the wireless-PLC mixed network in this work. To our knowledge, this is the first endeavor to portray the delay performance of mixed wireless-PLC by virtue of the SNC approach.

In this paper, we model the wireless-PLC mixed network as a two-stage tandem queue, over which the data is first relayed via a wireless link and then transmitted through a PLC system. The statistical characteristics of the service incremental process of both channels are analyzed. Furthermore, the delay upper bound for our wireless-PLC mixed system is evaluated via SNC in the SNR domain under any traffic model.

## 2. Network Model

We consider a wireless-PLC mixed cooperative communication system in this work. We assume that data are sent by the source to the relay first through a wireless channel. The relay is equipped with a hybrid capability for interfacing a wireless link with the PLC link. Then, data are transmitted from the relay to the destination over a PLC link. The wireless link between the source and relay is characterized by Rayleigh fading and additive white Gauss noise (AWGN). The instantaneous SNR over the wireless link is

$$\gamma_w(t) = \frac{P_w h_w(t)}{\sigma_w^2},\tag{1}$$

where $h_w$ incorporates the Rayleigh fading of the wireless link, which is given by an exponential random variable with a unit mean for mathematical tractability. $P_w$ is the transmit power, and $\sigma_w^2$ is the noise power of the wireless link.

The PLC link is modeled by the log-normal distributed channel gain and Bernoulli–Gaussian noise. Similarly, the instantaneous SNR of the PLC link can be written as

$$\gamma_p(t) = \frac{P_p h_p(t)}{n}, \tag{2}$$

where $h_p$ is the PLC channel gain whose amplitude is distributed as

$$f(a_{h_p}) = \frac{1}{a_{h_p}\sqrt{2\pi\sigma_{h_p}^2}} \exp\left(-\frac{(\ln a_{h_p} - \mu_{h_p})^2}{2\sigma_{h_p}^2}\right), \tag{3}$$

with $\ln a_{h_p}$ being a Gaussian random variable with mean $\mu_{h_p}$ and variance $\sigma_{h_p}^2$. For the Bernoulli–Gaussian noise, the probability distribution function (pdf) is written as

$$f(n) = (1-\rho)\mathcal{CN}(0,\sigma_g^2) + \rho\mathcal{CN}(0,\sigma_g^2 + \sigma_i^2). \tag{4}$$

Herein, $\mathcal{CN}(0,\sigma^2)$ is the complex Gaussian distribution with mean 0 and variance $\sigma^2$. $\rho$ is the probability of the impulsive component of the Bernoulli–Gaussian noise. The background noise power of the PLC link is $\sigma_g^2$. When considering both the background and impulsive noise, the total noise power is given as $\sigma_i^2 = \sigma_g^2 K$, where $K = \sigma_i^2/\sigma_g^2$ is the impulsive noise index.

With the above system and channel model, we proceed to analyze the delay bound for our wireless-PLC mixed network via SNC in the SNR domain.

## 3. Delay Analysis

Recall that the wireless-PLC mixed system in this paper is a discrete-time, fluid-flow queuing system and that the arrival and service process can be defined as $A(t)$ and $S(t)$. The arrival and service increments are denoted by $a$ and $s$, respectively. To make use of SNC in the SNR domain, we convert these two factors into the exponential domain as $\mathcal{A}(t) = e^{A(t)}$ and $\mathcal{S}(t) = e^{S(t)}$. We can easily obtain the probabilistic performance bounds of any non-negative random variable (RV) $X$ with the well-known Chernoff's bound $\Pr\{X(\tau,t) \geq x\} \leq x^{-\theta}\mathcal{M}_X(1+\theta,\tau,t)$, where $\mathcal{M}_X(\theta,\tau,t) = E[(X(\tau,t))^{\theta-1}]$ is the Mellin transform of RV $X$. With the Chernoff's inequality and the Mellin transform of the arrival and service incremental process, the delay bound can be easily calculated via SNC in the SNR domain ([16], Theorem 1).

In this work, we consider a Poisson arrival process for the data flow model. Provided that the arrival increment is a Poisson random variable with an average $\delta$, it can be equivalently characterized in the SNR domain as

$$\mathcal{M}_a(\theta) = \sum_{n=0}^{\infty} e^{n(\theta-1)}\frac{\delta^n}{n!}e^{-\delta} = e^{\delta(e^{\theta-1}-1)}. \tag{5}$$

Then, we proceed to evaluate the Mellin transform of the service incremental process of both the wireless link and PLC link. We denote $\gamma_p^1$ and $\gamma_p^2$ as the instantaneous SNRs. In the situation where impulsive noise is absent or present in the PLC link, respectively, we have

$$\gamma_p = \begin{cases} \gamma_p^1 = \frac{|h_p|^2 P_p}{\sigma_g^2}, & \text{only background noise,} \\ \gamma_p^2 = \frac{|h_p|^2 P_p}{\sigma_g^2(1+K)}, & \text{with impulsive noise.} \end{cases} \tag{6}$$

It is noteworthy that these two RVs in Equation (6) are also log-normal RVs, and their pdfs are given by

$$f_{\gamma_p^1}(\gamma) = \frac{1}{\gamma\sqrt{2\pi 4\sigma_{h_p}^2}} \exp\left(-\frac{(\ln\gamma - \ln\frac{P_p}{\sigma_g^2} - 2\mu_{h_p})^2}{8\sigma_{h_p}^2}\right),$$ (7)

$$f_{\gamma_p^2}(\gamma) = \frac{1}{\gamma\sqrt{2\pi 4\sigma_{h_p}^2}} \exp\left(-\frac{(\ln\gamma - \ln\frac{P_p}{\sigma_g^2(K+1)} - 2\mu_{h_p})^2}{8\sigma_{h_p}^2}\right).$$

With the above pdfs of the SNRs of the PLC link and the definition of the Mellin transform, we can further derive the Mellin transform of the service increments of the PLC link with Shannon capacity as

$$\mathcal{M}_{s_p}(\theta) = E\left[(1+\gamma)^{\frac{B(\theta-1)}{\ln 2}}\right] = \int_0^\infty (1+x)^{\frac{B(\theta-1)}{\ln 2}} f_{\gamma_p}(x)\mathbf{d}x,$$ (8)

where $B$ is the bandwidth of the wireless channel.

Inserting Equation (7) into Equation (8), the Mellin transform of the service incremental process of the PLC link is given by

$$\mathcal{M}_{s_p} = \begin{cases} \mathcal{M}_{s_p}^1 = \frac{1}{\sqrt{2\pi}2\sigma_{h_p}} \int_0^\infty (1+x)^{\frac{B(\theta-1)}{\ln 2}} x^{-1} \exp\left(-\frac{(\ln x - \ln\frac{P_p}{\sigma_g^2} - 2\mu_{h_p})^2}{8\sigma_{h_p}^2}\right)\mathbf{d}x, \\ \qquad\qquad \text{only background noise,} \\ \mathcal{M}_{s_p}^2 = \frac{1}{\sqrt{2\pi}2\sigma_{h_p}} \int_0^\infty (1+x)^{\frac{B(\theta-1)}{\ln 2}} x^{-1} \exp\left(-\frac{(\ln x - \ln\frac{P_p}{\sigma_g^2(K+1)} - 2\mu_{h_p})^2}{8\sigma_{h_p}^2}\right)\mathbf{d}x, \\ \qquad\qquad \text{with impulsive noise.} \end{cases}$$

With the independent property of the Mellin transform, the analytical average delay bound for the PLC link is expressed as

$$\mathcal{M}_{s_p} = (1-\rho)\mathcal{M}_{s_p}^1 + \rho\mathcal{M}_{s_p}^2.$$ (9)

Similarly, for the Rayleigh fading wireless link, the service incremental process can be calculated with the Shannon capacity. We transform the channel capacity into the SNR domain, and the Mellin transform of the service increment is

$$\mathcal{M}_{s_w}(\theta) = \exp\left(\frac{\sigma^2}{P_w}\right)\left(\frac{P_w}{\sigma^2}\right)^{\frac{B(\theta-1)}{\ln 2}} \Gamma\left(\frac{B(\theta-1)}{\ln 2}+1, \frac{\sigma^2}{P_w}\right).$$ (10)

**Proof.** Please refer to Appendix A. □

With the Mellin transform of the service increments of wireless and PLC links, we can easily calculate the delay bound for these two queues under any random traffic via SNC in the SNR domain. However, the total delay bound cannot be directly obtained by combining the two delays together, since we also need to consider that the output of wireless queue is the input of the PLC queue. Fortunately, the concatenation property of the SNC approach can be utilized to deal with this challenge. With the convolution operator of min-plus algebra, the two-stage tandem queue in this work can be equivalently viewed as a single queue, and the Mellin transform of the equivalent queue is

$$\mathcal{M}_{\mathcal{S}_{overall}}(\theta, \tau, t) = \mathcal{M}_{s_w \otimes s_p}(\theta, \tau, t)$$

$$\leq \sum_{u=\tau}^{t} \mathcal{M}_{s_w}(\theta, \tau, u) \cdot \mathcal{M}_{s_p}(\theta, u, t) \quad (11)$$

$$= \frac{\mathcal{M}_{s_p}^{t-\tau}(\theta) - \mathcal{M}_{s_w}^{t-\tau+1}(\theta)\mathcal{M}_{s_p}^{-1}(\theta)}{1 - \mathcal{M}_{s_w}(\theta)\mathcal{M}_{s_p}^{-1}(\theta)},$$

where $\otimes$ and $\oslash$ are the convolution and deconvolution operators that are defined in ([16], Equation (10)), and the bounds of the Mellin transform of the operators are given by ([16], Lemma 4). The last Equation in (11) follows from the rule of the sum of the geometric series. Then, the delay bound can be derived on the top of this page. Note that the first inequality is based on the property that, for a single queue with arrival $\mathcal{A}$ and service $\mathcal{S}$ in the SNR domain, the delay bound can be expressed as $W(t) = \inf\{u \geq 0 : \mathcal{A} \oslash \mathcal{S}(t, t+u) \leq 1\}$ ([16], Equation (16)).

$$\Pr\{W(t) > w\} \leq \Pr\{\mathcal{A} \oslash \mathcal{S}_{overall}(t, t+w) > 1\}$$

$$\leq \lim_{t \to \infty} \sum_{u=0}^{t} \mathcal{M}_{\mathcal{A}}(1+\theta, u, t)\mathcal{M}_{\mathcal{S}_{overall}}(1-\theta, u, t+w)$$

$$\leq \lim_{t \to \infty} \left[ \sum_{u=0}^{t} \mathcal{M}_{a}^{t-u}(1+\theta)\mathcal{M}_{s_p}^{t+w-u}(1-\theta) - \mathcal{M}_{s_p}^{-1}(1-\theta)\mathcal{M}_{a}^{t-u}(1+\theta) \right. \quad (12)$$

$$\left. \mathcal{M}_{s_w}^{t+w-u+1}(1-\theta) \right] / \left(1 - \mathcal{M}_{s_w}(1-\theta)\mathcal{M}_{s_p}^{-1}(1-\theta)\right)$$

$$= \frac{1}{1 - \mathcal{M}_{s_1}(1-\theta)\mathcal{M}_{s_2}^{-1}(1-\theta)} \times \left[ \frac{\mathcal{M}_{s_2}^{w}(1-\theta)}{1 - \mathcal{M}_{a}(1+\theta)\mathcal{M}_{s_2}(1-\theta)} \right.$$

$$\left. - \frac{\mathcal{M}_{s_2}^{-1}(1-\theta)\mathcal{M}_{s_1}^{w+1}(1-\theta)}{1 - \mathcal{M}_{a}(1+\theta)\mathcal{M}_{s_1}(1-\theta)} \right].$$

## 4. Numerical Results

In Figure 1, we investigate the impact of $\delta$ and $\frac{P_w}{\sigma^2}$ on the delay bound under different delay violation probabilities. Other system parameters are set as $B = \ln 2$ MHz, $\rho = 0.9$, $K = 20$ and $\frac{P_p}{\sigma_g^2} = 7$ dB. We observe that, with the increase of $\delta$, the delay bound becomes worse. This is due to the fact that more traffic arrivals mean more queue congestion, which will lead to a worse delay bound. Furthermore, for $\frac{P_p}{\sigma_g^2}$, a high value of SNR indicates a better transmission environment and a lower delay bound.

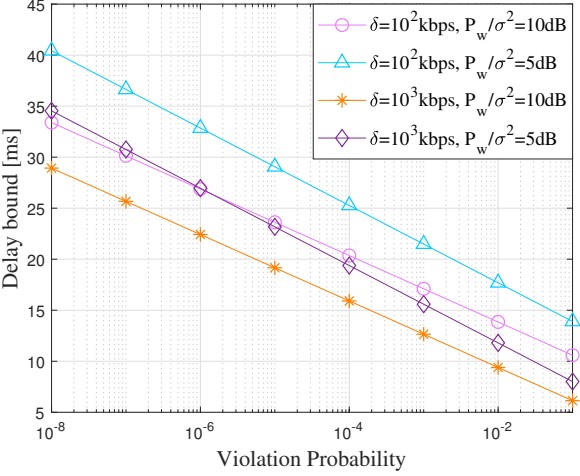

**Figure 1.** Delay versus violation probability under various data flows and $\frac{P_w}{\sigma^2}$.

In Figure 2, the parameter $\delta$ of the Poisson arrival flow is given in 100 kbps. We set the average SNR of the wireless link as $\frac{P_w}{\sigma^2} = 5$ dB. For the PLC link, the parameters are given as $\mu_{h_p} = 1$ and $\sigma^2_{h_p} = 1$. We conducted an experiment for the delay bound versus various average SNR of PLC link, under the different system parameters $K$ and $\rho$, where $K$ is the impulsive noise index and $\rho$ is the probability of the impulsive component of the noise. From the results, we observe that, with the increase of the value of $K$, the delay bound for the system becomes terrible. This phenomenon can be reasonably expressed from Equation (9). When parameter $K$ enlarges, the service capability of the PLC link under the circumstances of impulsive noise becomes weak as shown in Equation (6).

This also causes a decrease of the overall service capability according to Equation (9) and makes the delay bound larger. This expression also makes sense for the change of delay under different $\rho$. For other parameters, such as the data rate $\delta$ and average SNR $P_p/\sigma_g^2$, the impacts on the performance can be also evaluated via Figures 1 and 2. We observe that the delay bound increases with the increase of the data rate. This is reasonable since a higher data rate means more traffic congestion, which will result in a higher delay level. Similarly, a larger value of $P_p/\sigma_g^2$ indicates a better transmission environment and a lower delay bound.

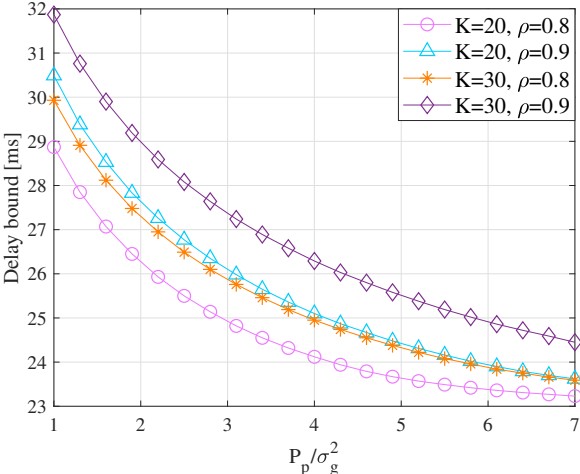

**Figure 2.** Delay versus $\frac{P_p}{\sigma_g^2}$ under various PLC channels.

## 5. Conclusions

In this paper, we modeled the wireless-PLC mixed network as a two-stage tandem queue, wherein the wireless link was Rayleigh fading and the PLC R-D link was modeled by the log-normal distributed channel gain and Bernoulli–Gaussian noise. SNC in the SNR domain was exploited to derive the delay bounds for the network. Furthermore, we provided numerical results to demonstrate the impacts of various factors of the network. Our work here provides insights regarding the network deployment and flow control for wireless-PLC mixed networks under different delay requirements.

**Author Contributions:** Conceptualization, M.M. and Q.Y.; methodology, M.M.; software, Z.L.; validation, Z.L., H.H. and L.P.; formal analysis, Z.L.; investigation, Z.L.; resources, H.H.; data curation, L.P.; writing—original draft preparation, M.M.; writing—review and editing, Q.Y.; visualization, Z.L.; supervision, Z.L.; project administration, Z.L.; funding acquisition, Z.L. All authors have read and agreed to the published version of the manuscript.

**Funding:** This work was supported by The Laboratory Open Fund of Beijing Smart-chip Microelectronics Technology Co., Ltd.

**Institutional Review Board Statement:** Not applicable.

**Informed Consent Statement:** Not applicable.

**Data Availability Statement:** Not applicable.

**Acknowledgments:** The authors wish to thank the anonymous reviewers for their help comments.

**Conflicts of Interest:** The authors declares no conflict of interest.

## Appendix A

Proof of the derivation of Equation (10): with the definition of the Mellin transform, we have

$$
\mathcal{M}_{s_w}(\theta) = E\left[ (1 + \gamma_w)^{\frac{B(\theta-1)}{\ln 2}} \right]
$$

$$
= \int_0^{\infty} \left(1 + \frac{P_w x}{\sigma^2}\right)^{\frac{B(\theta-1)}{\ln 2}} \exp(-x)\mathbf{d}x
$$

$$
= \int_1^{\infty} y^{\frac{B(\theta-1)}{\ln 2}} \exp\left(\frac{\sigma^2}{P_w}(1-y)\right)\mathbf{d}\left(\frac{\sigma^2}{P_w}(y-1)\right)
$$

$$
= \frac{\sigma^2}{P_w} \exp\left(\frac{\sigma^2}{P_w}\right) \int_1^{\infty} y^{\frac{B(\theta-1)}{\ln 2}} \exp\left(-\frac{\sigma^2 y}{P_w}\right)\mathbf{d}y
$$

$$
= \frac{\sigma^2}{P_w} \exp\left(\frac{\sigma^2}{P_w}\right) \int_{\frac{\sigma^2}{P_w}}^{\infty} \left(\frac{P_w}{\sigma^2}\right)^{\frac{B(\theta-1)}{\ln 2}} z^{\frac{B(\theta-1)}{\ln 2}} \times
$$

$$
\exp(-z)\mathbf{d}\left(\frac{P_w z}{\sigma^2}\right)
$$

$$
= \exp\left(\frac{\sigma^2}{P_w}\right) \left(\frac{P_w}{\sigma^2}\right)^{\frac{B(\theta-1)}{\ln 2}} \int_{\frac{\sigma^2}{P_w}}^{\infty} z^{\frac{B(\theta-1)}{\ln 2}} \exp(-z)\mathbf{d}z
$$

$$
= \exp\left(\frac{\sigma^2}{P_w}\right) \left(\frac{P_w}{\sigma^2}\right)^{\frac{B(\theta-1)}{\ln 2}} \Gamma\left(\frac{B(\theta-1)}{\ln 2} + 1, \frac{\sigma^2}{P_w}\right),
$$

where in the last step, we take advantage of the definition of the incomplete Gamma function $\Gamma(s,y) = \int_y^{\infty} x^{s-1} \exp(-x)\mathbf{d}x$.

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
