# Peer review of "Stochastic Network Calculus-Aided Delay Analysis of Wireless-Power Line Mixed Networks"

_electronics, doi:10.3390/electronics11091326_

Round 1

Reviewer 1 Report

In this manuscript, the authors have modelled a wireless PLC mixed network using stochastic-network-calculus (SNC) based approach. The derivation to calculate delay bound has been presented, and numerical results are presented to show the impact of violation probability under various data flow and SNR. There are a few issues which need to be addressed,

  1. Why Log-normal distributed channel gain and Bernoulli-
    Gaussian noise have been chosen to model the PLC link? Also comment on why Poisson arrival process is used for data flow model?
  2. In the numerical results section, define the parameters K and ρ. Also, please comment on other system parameters which can impact the delay bound.
  3. How the delay bound can be reduced further? And, what optimization steps are required to obtain the ideal system parameters to achieve the lower delay bound? Please describe.

Reviewer 2 Report

Please see the attached Comment to the Authors.

Round 2

Reviewer 2 Report

Authors have adequately improved the manuscript taking into account the reviewer’s comments. The revised manuscript can be accepted in current form.